

# Global and regional carbon budget 2015–2020 inferred from OCO-2 based on an ensemble Kalman filter coupled with GEOS-Chem

Yawen Kong[1], Bo Zheng[2,3], Qiang Zhang[1], Kebin He[3,4]

[1]Ministry of Education Key Laboratory for Earth System Modeling, Department of Earth System Science, Tsinghua
University, Beijing 100084, China
[2]Institute of Environment and Ecology, Tsinghua Shenzhen International Graduate School, Tsinghua University, Shenzhen
518055, China
[3]State Environmental Protection Key Laboratory of Sources and Control of Air Pollution Complex, Beijing 100084, China
[4]State Key Joint Laboratory of Environment Simulation and Pollution Control, School of Environment, Tsinghua University,
Beijing 100084, China

*Correspondence*: Bo Zheng (bozheng@sz.tsinghua.edu.cn)

**Abstract.** Understanding carbon sources and sinks across the Earth's surface is fundamental in climate science and policy;
thus, these topics have been extensively studied but have yet to be fully resolved and are associated with massive debate
regarding the sign and magnitude of the carbon budget from global to regional scales. Developing new models and estimates
based on state-of-the-art algorithms and data constraints can provide valuable knowledge and contribute to a final ensemble
model in which various optimal carbon budget estimates are integrated, such as the annual Global Carbon Budget paper.
Here, we develop a new atmospheric inversion system based on the four-dimensional local ensemble transform Kalman filter
(4D-LETKF) coupled with the GEOS-Chem global transport model to infer surface-to-atmosphere net carbon fluxes from
Orbiting Carbon Observatory-2 (OCO-2) V10r $XCO_2$ retrievals. The 4D-LETKF algorithm is adapted to an OCO-2-based
global carbon inversion system for the first time in this work. On average, the mean annual terrestrial and oceanic fluxes
between 2015 and 2020 are estimated as −2.02 GtC yr$^{-1}$ and −2.34 GtC yr$^{-1}$, respectively, compensating for 21% and 24%,
respectively, of global fossil $CO_2$ emissions (9.80 GtC yr$^{-1}$). Our inversion results agree with the $CO_2$ atmospheric growth
rates reported by the National Oceanic and Atmospheric Administration (NOAA) and reduce the modelled $CO_2$
concentration biases relative to the prior fluxes against surface and aircraft measurements. Our inversion-based carbon fluxes
are broadly consistent with those provided by other global atmospheric inversion models, although discrepancies still occur
in the land-ocean flux partitioning schemes and seasonal flux amplitudes over boreal and tropical regions, possibly due to the
sparse observational constraints of the OCO-2 satellite and the divergent prior fluxes used in different inversion models.
Four sensitivity experiments are performed herein to vary the prior fluxes and uncertainties in our inversion system,
suggesting that regions that lack OCO-2 coverage are sensitive to the priors, especially over the tropics and high latitudes. In
the further development of our inversion system, we will optimize the data-assimilation configuration to fully utilize current
observations and increase the spatial and seasonal representativeness of the prior fluxes over regions that lack observations.



## 1 Introduction

The atmospheric concentration of carbon dioxide ($CO_2$) reached 414.7 parts per million (ppm) in 2021, rising 49% above the
preindustrial level (Friedlingstein et al., 2021); this increased $CO_2$ continuously enhances the greenhouse effect and global
warming. To predict and mitigate climate change, it is of critical importance to understand how much $CO_2$ is released and
absorbed by human and natural systems, where these exchanges occur, and how these carbon fluxes respond to
anthropogenic and natural forcings (Canadell et al., 2021). Atmospheric $CO_2$ measurements have indicated that on average,
half of the $CO_2$ emitted by humans from fossil fuels and land-use changes globally is taken up by the oceans and land each
year (Ciais et al., 2019), and the spatiotemporal distributions of global and regional carbon budget must be further
reconstructed and analysed using increasingly sophisticated bottom-up and top-down approaches.

Top-down methods, unlike bottom-up methods in which carbon sources and sinks are simulated by process-based models,
infer carbon fluxes from observed spatiotemporal $CO_2$ concentration gradients within each carbon reservoir (Gurney et al.,
2002). Surface fluxes are estimated by conducting data inversions using atmospheric transport models in a Bayesian
framework to correct prior carbon fluxes to match measured $CO_2$ concentrations within the error structures of the priors and
observations (Ciais et al., 2010). Various global $CO_2$ atmospheric inversion systems have been developed over the past
decades, and these systems differ in their associated transport models, assimilated observations, and inversion algorithms.
Different inversion systems tend to estimate consistent global total net carbon fluxes due to the carbon mass balance being
constrained by global atmospheric measurements; however, large discrepancies have been reported at fine spatiotemporal
scales (e.g., monthly zonal averages), and these discrepancies have urged scientists to organize community-wide inverse
model intercomparisons to identify and mitigate gaps in the understanding of carbon cycle dynamics. Such international
collaboration efforts include the Atmospheric Tracer Transport Model Intercomparison 3 (TransCom 3) (Gurney et al., 2003),
the REgional Carbon Cycle Assessment and Processes (RECCAP) (Peylin et al., 2013; Ciais et al., 2022), the Orbiting
Carbon Observatory-2 Model Intercomparison Project (OCO-2 MIP) (Crowell et al., 2019), and many other inverse model
intercomparison studies (Chevallier et al., 2014; Houweling et al., 2015; Thompson et al., 2016).

Global carbon inversion estimates tend to converge with these preceding intercomparison projects, although discrepancies
and uncertainties still exist and persist today (Basu et al., 2018; Gaubert et al., 2019; Bastos et al., 2020). Further
atmospheric inversion improvements could be obtained through technological advancements of the observations, data
assimilation techniques, atmospheric transport models, prior fluxes, and associated error statistics. The recently accelerated
expansion of carbon measurement networks (e.g., ground-, aircraft-, and space-based platforms) has enhanced our
capabilities to constrain and evaluate atmospheric inversion models. Most remarkably, the continuous improvements in $CO_2$
column retrievals from satellites such as the Orbiting Carbon Observatory-2 (OCO-2) (Eldering et al., 2017) have
substantially promoted satellite-based carbon inversion estimates, which are now comparable to surface measurement-based
inversions in terms of their credibility (Chevallier et al., 2019). Further developments associated with the fundamental roles
of carbon atmospheric inversions in climate science and policymaking as well as the current large model spread are still





urgently required. The utilization of rapidly evolving satellite retrievals in combination with the latest transport models and data assimilation techniques represents a future direction to improve our understanding of global and regional carbon cycle.

In this study, we develop a new global $CO_2$ atmospheric inversion system based on the four-dimensional local ensemble transform Kalman filter (4D-LETKF) coupled with the GEOS-Chem global transport model to estimate surface-to-
atmosphere net carbon fluxes from 2015 to 2020. The 4D-LETKF is a variant of the ensemble Kalman filter (Hunt et al., 2007) and has been applied in various atmospheric data assimilation studies demonstrating its efficiency and accuracy (Houtekamer and Zhang, 2016). Liu et al. (2019) analysed the feasibility of using the 4D-LETKF algorithm in global carbon inversions through an observing system simulation experiment based on pseudo-observations. Here, for what is, to our knowledge, the first time, we adapt the 4D-LETKF algorithm to establish a global carbon inversion system that is
constrained by realistic space-based retrievals of the column-averaged dry air mole fraction of $CO_2$ ($XCO_2$). The latest OCO-2 V10r bias-corrected $XCO_2$ retrievals (OCO-2 Science Team, 2020) are assimilated into our inversion system. We conduct a comprehensive evaluation of our carbon inversion results through 1) an independent evaluation against surface- and aircraft-derived $CO_2$ measurements by latitude, 2) four sensitivity experiments with varied prior fluxes, error structures, and assimilation window length, and 3) comprehensive comparisons with other state-of-the-art inversion model estimates to
investigate both the consistencies and inconsistencies among the models and explore their possible drivers. Because our inversion system is built upon a new inversion algorithm and the latest OCO-2 retrievals, it can contribute to an ensemble of existing global $CO_2$ inversions and help constrain carbon inversion model spread and reduce uncertainties.

In the remainder of this paper, the utilized configurations, models, data inputs, and observation-based evaluations associated with the inversion system are described in Sect. 2, and the $CO_2$ budget inversion estimates are analysed at the global and
regional scales in Sect. 3. The sensitivity inversion results are presented and discussed in Sect. 4, and Sect. 5 contains a summary of the findings obtained in this study.

## 2 Data and methods

### 2.1 Carbon flux inversion system

We developed a Bayesian atmospheric inversion system (Fig. 1) to infer daily gridded surface carbon fluxes (excluding
fossil fuel and biomass burning emissions, which were prescribed) from OCO-2 $XCO_2$ retrievals. This system was built upon the GEOS-Chem global transport model coupled with the 4D-LETKF algorithm (Hunt et al., 2007) and is analogous to identifying a weight that linearly combines the ensemble members of carbon fluxes to obtain best-fitting $CO_2$ observations. Our system assimilates OCO-2 $XCO_2$ on an ongoing basis and optimizes carbon fluxes on the first day of each assimilation window by minimizing a cost function as follows (Eq. (1)):

$$J(\mathbf{x}) = \left(\mathbf{x} - \mathbf{x}^b\right)^\mathrm{T} \mathbf{B}^{-1} \left(\mathbf{x} - \mathbf{x}^b\right) + \left(\mathbf{y} - H(\mathbf{x})\right)^\mathrm{T} \mathbf{R}^{-1} \left(\mathbf{y} - H(\mathbf{x})\right) \tag{1}$$





where $x$ is a control vector consisting of variables to be optimized (i.e., the scale factors of the surface fluxes in each grid cell), $x^b$ is a prior guess corresponding to the control vector $x$ with errors represented by a covariance matrix $\mathbf{B}$, $y$ is an observation vector that gathers the OCO-2 $XCO_2$ retrievals, $\mathbf{R}$ represents the error covariance matrix, and $H$ is the observation operator, which calculates the OCO-2-equivalent $XCO_2$ value from the GEOS-Chem simulations, OCO-2 $XCO_2$
prior, and column averaging kernel. The cost function $J(x)$ measures the differential surface carbon fluxes between the prior ($x^b$) and optimized ($x$) estimates plus the difference in the $XCO_2$ fields between the OCO-2 observations ($y$) and GEOS-Chem simulations ($H(x)$); these two terms are weighted using the prior errors ($\mathbf{B}$) and observation errors ($\mathbf{R}$), respectively.

In each data assimilation window, the control vector ($x$) is optimized through Eqs. (2)–(5) as follows (Hunt et al., 2007):

$$\overline{x}^a = \overline{x}^b + \mathbf{X}^b \overline{w}^a \tag{2}$$

$$\mathbf{X}^a = \mathbf{X}^b \left[ (k-1)\tilde{\mathbf{P}}^a \right]^{1/2} \tag{3}$$

$$\overline{w}^a = \tilde{\mathbf{P}}^a \left( \mathbf{Y}^b \right)^{\mathrm{T}} \mathbf{R}^{-1} \left( y^o - \overline{y}^b \right) \tag{4}$$

$$\tilde{\mathbf{P}}^a = \left[ (k-1)\mathbf{I} + \left( \mathbf{Y}^b \right)^{\mathrm{T}} \mathbf{R}^{-1} \mathbf{Y}^b \right]^{-1} \tag{5}$$

where $a$ and $b$ represent the posterior and prior state, respectively, $k$ represents the ensemble size (i.e., 24), $\overline{x}$ is the ensemble mean of the control vector, $\mathbf{X}$ is the ensemble perturbation matrix whose $i$th column represents $x^{(i)} - \overline{x}$ {$i = 1,2,…,k$}, $y^o$
contains the assimilated OCO-2 $XCO_2$ within the assimilation window and localization length, $\overline{y}^b$ is the mean of a prior $XCO_2$ field averaged over $y^{b(i)} = H(x^{b(i)})$ {$i = 1,2,…,k$} simulations, $\mathbf{Y}^b$ is the ensemble perturbation matrix whose $i$th column represents $y^{b(i)} - \overline{y}^b$ {$i = 1,2,…,k$}, $\overline{w}^a$ is a weight vector, $\tilde{\mathbf{P}}^a$ is the analysis covariance matrix, and $\mathbf{I}$ is the identity matrix.

The ensemble mean $\overline{x}^b$ is calculated by obtaining the average optimized result from the two previous time steps and a fixed value of one (Peters et al., 2007), thus propagating the optimized information from two previous steps to the current state and
representing a moving average smoothing technique that suppresses variations in $x^b$ over time. The error structure of $x^b$ was constructed based on a normal distribution with the standard deviation of 3.0 within a spatial correlation length of 2000 km, which approximates the variance in the ensemble members $x^{b(i)}$ {$i = 1, 2, …, 24$} from their mean. The spatial correlation of the prior flux errors between ocean and land is set zero in our inversion. The term $\overline{w}^a$ is a weight vector that specifies the linear combinations of ensemble perturbations ($\mathbf{X}^b$) that are added to the prior mean ($\overline{x}^b$) to estimate the posterior mean ($\overline{x}^a$).
The ensemble mean of $\overline{x}^a$ is then used to update the carbon fluxes at the current assimilation window, thus driving another GEOS-Chem simulation to generate the initial $CO_2$ concentration fields for the next assimilation cycle.

Our inversion system configuration is summarized in Table 1. The assimilation window was set to 7 days based on the inversion system configurations of Zhang et al. (2015), Liu et al. (2019), and Jiang et al. (2021). The carbon fluxes



representing the first day of each window are optimized based on the link between the modelled $XCO_2$ and observed $XCO_2$
within the assimilation window. The GEOS-Chem model and prior fluxes are described in Sect. 2.2 below, and Sect. 2.3
describes how the OCO-2 $XCO_2$ observations and their uncertainties were assimilated. In Sect. 2.4, we designed four
sensitivity inversions to vary the prior fluxes and prior uncertainty statistics and investigate their influence on the inversion
results. The procedure followed to evaluate the posterior fluxes and independent observations are presented in Sect. 2.5.

## 2.2 Transport model and carbon fluxes

GEOS-Chem is a global 3-D chemical transport model (Bey et al., 2001; https://geos-chem.seas.harvard.edu/) driven by
meteorological fields obtained from the Goddard Earth Observing System (GEOS) of the National Aeronautics and Space
Administration (NASA) Global Modelling and Assimilation Office. GEOS-Chem has been applied to develop global carbon
inversion systems by several research groups worldwide (Feng et al., 2009; Deng et al., 2014; Liu et al., 2021), and the
resulting systems vary according to their model versions, data assimilation methods, and utilized prior fluxes. Here, we used
GEOS-Chem v12.2.1 in our inversion system to simulate the global $CO_2$ transport and relate surface carbon fluxes to
observed atmospheric $CO_2$ gradients at a horizontal resolution of 4° latitude × 5° longitude, driven by GEOS-FP
meteorology data. Such a spatial resolution is sufficient to capture large-scale atmospheric $CO_2$ transport along with the
associated spatiotemporal variability and can achieve a balance between ensemble simulations and computational costs.

We distinguished among four $CO_2$ flux categories in the GEOS-Chem model, including fossil fuel fluxes, biomass burning
fluxes, ocean fluxes, and terrestrial biospheric fluxes. The fossil fuel emissions from land and international bunker sources
were derived from the Open-source Data Inventory for Anthropogenic $CO_2$ (ODIAC, version 2020) dataset (Oda et al., 2018)
for 2014–2019, and we downscaled the dataset from the monthly to hour scale based on temporal scaling factors obtained
from the Temporal Improvements for Modeling Emissions by Scaling (TIMES) database (Nassar et al. 2013). The 2020
emissions were estimated by extrapolating daily 2019 emissions based on the emission growth rates from 2019 to 2020
derived from the Carbon Monitor project (Liu et al., 2020, https://carbonmonitor.org/). The biomass burning emissions were
obtained from the Global Fire Emissions Database (GFED) 4.1s (van der Werf et al., 2017) from 2014 to 2020; this database
provides monthly emissions of different fire types and daily and 3-hourly temporal profiles. These monthly biomass burning
emissions were downscaled to 3-hourly fluxes. Ocean-atmosphere $CO_2$ fluxes on a 3-hourly basis were obtained from the
$p$CO$_2$-Clim prior of the CarbonTracker version CT2019B (CT2019B) (Takahashi et al. 2009; Jacobson et al., 2020). The 3-
hour terrestrial biospheric fluxes were derived from the Simple Biosphere Model, Verison 4.2 (SiB4) global hourly dataset
(Haynes et al., 2021). We halved the gridded terrestrial biospheric fluxes to dampen the seasonal cycle and then integrated
annual fluxes as zero over land, thus implying that the spatial-temporal variabilities in inferred terrestrial fluxes from our
inversion system were mainly determined by the assimilated observations. The 2018 prior ocean and terrestrial biospheric
fluxes were used for the 2019 and 2020 inversions because CT2019B and SiB4 data were available up to 2018 at present.



## 2.3 Assimilated OCO-2 observations

OCO-2 is the first dedicated $CO_2$-monitoring satellite designed by NASA; this satellite was launched in July 2014 (Eldering et al., 2017). It flies in a sun-synchronous, near-polar orbit 705 km above the Earth's surface with a repeat cycle of 16 days and a local overpass time of approximately 1:30 pm. OCO-2 collects 8 adjacent cross-track samples every 0.333 seconds (24 samples per second) at a spatial resolution of 1.29 km × 2.25 km for each footprint at nadir. We assimilated the OCO-2 Level 2 bias-corrected $XCO_2$ retrievals, retrospective processing V10r (OCO-2 Science Team, 2020) in our inversion system. Figure 2 presents the spatial and seasonal distributions of valid OCO-2 V10r $XCO_2$ retrievals over the 4° × 5° GEOS-Chem grid cells between 2015 and 2020.

The high-density OCO-2 $XCO_2$ retrievals were preprocessed to generate 1-s and 10-s averages before being assimilated because the retrieval errors were closely correlated both temporally and spatially (Crowell et al., 2019). First, the "good" retrievals in the OCO-2 Lite files were selected according to the "xco2_quality_flag" variable and filtered to remove outliers in each orbit using the "3 times the standard deviation" rule. That is, $XCO_2$ values whose differences from their adjacent soundings deviated from the mean by more than three times the standard deviation were filtered out and not used in the subsequent data assimilation process. Then, 1-s averages were computed from the selected good retrievals across each 1-s span along the OCO-2 tracks using the method described by Crowell et al. (2019). The inverse error variance obtained for each $XCO_2$ retrieval was used to calculate a weighted average for all related variables with the uncertainty represented by an average uncertainty of the adopted single soundings. Finally, 10-s averages were computed across each 10-s span (approximately corresponding to a ground track 70 km in length) by weighting the 1-s averages by their inverse variance values. The uncertainty of these 10-s averages was estimated as an average uncertainty for the adopted 1-s averages inflated by factors of 1.8 and 1.4 over lands and oceans, respectively; the results thus accounted for the representation errors that arose due to the mismatches between the GEOS-Chem model and assimilated OCO-2 observation resolutions. The 10-s averages were then assimilated to our inversion system while assuming independence among the different 10-s spans.

## 2.4 Sensitivity inversion experiments

We performed four inversion sensitivity experiments using different prior fluxes, uncertainty configurations, and assimilation window length to investigate the influence of these factors on the resulting carbon inversions (Table 2). Based on the reference inversion, we reduced and increased the standard deviations of the normal distributions used to represent the error structures of $x^b$ in sensitivity experiments no. 1 (S_exp1) and no. 2 (S_exp2), respectively, to quantify the influence of the ensemble spread of $x^b$ on the carbon inversion results. In sensitivity experiment no. 3 (S_exp3), the prior terrestrial biospheric fluxes from CT2019B were used; these fluxes are based on the Carnegie-Ames Stanford Approach (CASA) biogeochemical model, and the other configurations remained the same as those used in the reference inversion. The comparison between S_exp3 and our reference inversion illustrates the impact of different prior terrestrial biospheric fluxes on the inversion results. In sensitivity experiment no. 4 (S_exp4), the data assimilation window length was doubled to 14



days compared to the reference inversion, which tends to constrain fluxes based on more OCO-2 observations in each assimilation window. All four sensitivity inversions were performed considering the period from September 2014 to December 2015, thus providing inversion results 2015 for use and comparison in our analysis.

**2.5 Evaluation of posterior fluxes**

We compared the GEOS-Chem-modelled dry air mole fractions of $CO_2$ based on posterior fluxes with independent surface and aircraft measurements to evaluate the posterior fluxes. These measurement data were not assimilated into our inversion system. Such evaluation methods have been widely used to evaluate global carbon budget estimates inferred from atmospheric inversion (Chevallier et al., 2019; Crowell et al., 2019). The evaluation observation datasets were obtained from

the $CO_2$ GLOBALVIEWplus v7.0 ObsPack database (Schuldt et al., 2021), which is maintained by the Earth System Research Laboratory (ESRL) of the National Oceanic and Atmospheric Administration (NOAA) (https://www.esrl.noaa.gov/gmd/ccgg/obspack/). The ObsPack framework (Masarie et al., 2014) archives direct atmospheric greenhouse gas measurements from different laboratories to support carbon cycle modelling research. We collected flask sample measurements from 52 stations (Table S1) at altitudes lower than 3000 m and aircraft measurements from 3

programs (i.e., ABOVE, ACT, and TOM, please see Table S2) between 2015 and 2020. To perform the evaluation, GEOS-Chem model-simulated $CO_2$ concentrations were sampled at the locations and times corresponding to the observation data points to calculate the multiannual mean bias and root mean square error (RMSE) values by season and by latitude band.

In addition, we collected surface carbon flux estimates from different atmospheric inversion models, including NOAA's CT2019B (Jacobson et al., 2020), the Copernicus Atmosphere Monitoring Service (CAMS) model versions v20r2 and v20r3

(Chevallier, et al., 2005), Jena CarboScope version sEXTocNEET_v2021 (Rödenbeck et al., 2018), and the Carbon Monitoring System Flux (CMS-Flux) (Liu et al., 2021). These carbon budget products were built upon different atmospheric inversion frameworks that vary with different transport models, observation constraints, and assimilation techniques. We performed comprehensive comparisons at both the global and regional scales to evaluate our inversion estimates.

**3 Results**

**3.1 Global carbon budget**

The annual global carbon budgets derived using our inversion system are shown for 2015–2020 in Table 3. The mean annual terrestrial flux—the sum of the net ecosystem exchange ($-3.91$ GtC yr$^{-1}$) and fire (1.88 GtC yr$^{-1}$) fluxes—was estimated as $-2.02$ GtC yr$^{-1}$, and the mean annual oceanic flux was estimated as $-2.34$ GtC yr$^{-1}$. On average, the terrestrial and oceanic fluxes compensated for 21% and 24%, respectively, of the global fossil $CO_2$ emissions (9.80 GtC yr$^{-1}$), with the remaining

55% of fossil $CO_2$ emissions (5.44 GtC yr$^{-1}$) remaining in the atmosphere. Our inversion results agreed with NOAA's surface measurement-based atmospheric $CO_2$ growth rates (https://gml.noaa.gov/ccgg/trends/gl_gr.html); this source



reported average annual growth of 5.39 GtC yr$^{-1}$ from 2015 to 2020 based on the conversion factor of 2.124 GtC ppm$^{-1}$ (Friedlingstein et al., 2021). The derived bias of 0.05 GtC yr$^{-1}$ was slightly lower than the bias range of the atmospheric inversion models (0.06–0.17 GtC yr$^{-1}$) that participated in the Global Carbon Budget 2021 (GCB2021) project

(Friedlingstein et al., 2021). The broad consistency between our inversion results and the atmospheric $CO_2$ growth rate from measurement suggests that the net atmosphere-surface exchange of $CO_2$ was well constrained by our inversion system.

The global carbon budget partitioning results are shown in Fig. 3, including our reference inversion results, other state-of-the-art atmospheric inversion estimates, and the ensemble estimates from GCB2021 (riverine flux-adjusted) for 2015–2018, the common period when all of these data were available. The integrated land (with fossil $CO_2$ emissions) and ocean fluxes

were scattered around the purple diagonal line denoting the atmospheric growth rate in Fig. 3a, suggesting that the global-scale $CO_2$ fluxes were conserved and well-constrained in all of the considered inversion models, although these models assimilated different $CO_2$ observations using various strategies. Our inversion system used a relatively large prior for land fluxes involving a combination of prescribed biomass burning emissions (~1.80 GtC yr$^{-1}$) and annually zero terrestrial biospheric fluxes, while the other inversion models used annually zero or negative prior natural land fluxes (Fig. 3b). Despite

the large prior land fluxes used by our model (denoted by the red open circles shown in Figs. 3a and 3b), our inversion system successfully corrected the global fluxes to match the atmospheric $CO_2$ growth rates (the red solid circles in Figs. 3a and 3b). The major discrepancies derived from different inverse models involved the partitioning scheme between land and ocean fluxes. Our inversion results, as well as CAMS and Jena, estimated smaller land fluxes and ocean uptakes than CMS-Flux and CT2019B. GCB2021 is comparable to our inversion estimates but presents a large budget imbalance (−0.63 GtC

yr$^{-1}$ averaged between 2015 and 2018) due to model deficiencies (Friedlingstein et al., 2021); this imbalance explained why the purple circles representing the GCB2021 estimates were not located on the purple lines in Figs. 3a and 3b.

The differences among inversion-based global carbon budget estimates are mainly attributed to the natural components, not fossil fuel emissions, as illustrated by the large spread of natural fluxes (without the prescribed fossil $CO_2$ emissions) in Fig. 3b. This finding differs from previous intercomparison studies in which global atmospheric $CO_2$ inverse models were shown

to disagree on fossil fuel priors (Gaubert et al., 2019). All of the atmospheric inversion models shown in Fig. 3 adopted consistent fossil $CO_2$ priors with an annual average of 9.7–10.0 GtC yr$^{-1}$ from 2015–2018, benefiting from the community efforts to constrain uncertainties associated with fossil fuel emissions and converge on global total carbon sinks. However, the land-ocean partitioning schemes of natural fluxes are much more uncertain and can reflect spreads up to 1.6 GtC yr$^{-1}$ (Fig. 3b), comparable with the uncertainties associated with the land-ocean partitioning scheme due to transport model

differences reported by Basu et al. (2018). Moreover, we found that the global oceanic fluxes seemed to be largely unchanged relative to the ocean priors used in different inverse models (Fig. 3b), likely due to the weak observational constraints over oceans. The usage of prior ocean fluxes that differ by 1.1 GtC yr$^{-1}$ among inverse models thus plays an important role in determining the land-ocean partitioning schemes of global fluxes in atmospheric inversion results.



## 3.2 Regional carbon budget

### 3.2.1 Latitudinal distribution of fluxes

The partitioning divides between the northern extratropics (23°N–90°N) (NET) and the tropics (23°S–23°N) (T) + southern extratropics (90°S–23°S) (SET) are illustrated in Figs. 3c and 3d. The integrated fluxes from NET and T+SET are anticorrelated and scattered around the global atmospheric growth rate of $CO_2$ minus the fossil $CO_2$ emissions listed in GCB2021 (the purple diagonal line in Fig. 3c), suggesting that the global total natural fluxes were well-constrained by
atmospheric inversion. Our inversion results suggest that NET and T+SET represent average natural fluxes of −3.5 GtC yr$^{-1}$ and −0.9 GtC yr$^{-1}$, respectively, between 2015 and 2018, both of which lie within the ensemble of different inversion model estimates (Figs. 3c and 3d). Land fluxes dominate over ocean fluxes in NET, which are estimated as −2.6 GtC yr$^{-1}$ and −0.9 GtC yr$^{-1}$, respectively, on average between 2015 and 2018. However, in T+SET, ocean fluxes (−1.4 GtC yr$^{-1}$) dominate over land fluxes (0.6 GtC yr$^{-1}$), because the area of ocean is much larger than land in this region. The differences among inversion
model estimates could be attributed to land-ocean partitioning by latitude, although the discrepancies in posterior fluxes derived are largely reduced compared to the prior fluxes after assimilating the OCO-2 $CO_2$ observations. GCB2021 tended to give larger land sinks than all of the atmospheric inversion models except for CAMS (Fig. 3d), although the large budget imbalance of GCB2021 complicated the interpretation of these large flux discrepancies. The disagreements among multiple inversion models over latitude indicated the existence of substantial uncertainties in the regional carbon budget estimates.

### 3.2.2 Regional distribution of fluxes

Figure 4 presents the spatial distribution of the natural fluxes derived from our reference inversion model, including both prior and posterior annual average fluxes between 2015 and 2020. We estimated a net land carbon flux of −2.4 GtC yr$^{-1}$ between 2015 and 2020 over the Northern Hemisphere; this value was slightly larger than the −2.1±0.5 GtC yr$^{-1}$ estimate obtained from 2000–2010 by Ciais et al. (2019) based on a two-box atmospheric inversion model. Figure 4 shows that large
carbon sinks are located in the northern forests and woodlands over the eastern USA, Asia, and Europe, as well as in the tropical evergreen forests over South America and Africa (Fig. 4c). Since the prior biospheric annual flux was integrated as zero over land globally (Fig. 4a), the spatial distribution of the posterior carbon sinks was reconstructed only by the assimilated OCO-2 XCO$_2$ through atmospheric inversion. When the biomass burning fluxes were added, as was prescribed in our inversion system (Fig. 4b), we observed large flux gradients over South America, southern Africa, and the Eurasia
boreal region in the posterior fluxes (Fig. 4d) due to extensive savanna and forest fires (van der Werf et al., 2017).

Over the 11 TransCom land regions (Fig. 5), our inversion results were broadly consistent with the other atmospheric inversion products, although we did observe an unsurprising lack of agreement due to large uncertainties in regional flux estimates. Northern Africa, Southern Africa, and South American Tropical all represent net carbon sources due to the substantial fire emissions, especially from forest fires, in these regions. Large net carbon uptakes occur over North America,
Eurasia, and Europe, where our inversion model estimated annual average fluxes of −0.3 GtC yr$^{-1}$, −1.4 GtC yr$^{-1}$, and −0.9





GtC yr$^{-1}$, respectively; further, these estimates broadly agreed with the ensemble of surface observation-based atmospheric CO$_2$ inversions derived between 2001 and 2004 (Peylin et al., 2013), which provided flux values of −0.7±0.5 GtC yr$^{-1}$, −1.1±0.4 GtC yr$^{-1}$, and −0.4±0.5 GtC yr$^{-1}$, respectively, over these three regions. Our inversion model seemed to estimate slightly lower fluxes over the boreal and temperate regions in North America than the other inversion models. The OCO-2 land observations were limited to lower latitudes during fall and winter in the Northern Hemisphere (Fig. 2), and the retrieval biases increased with the solar and satellite zenith angles (O'Dell et al., 2018). We would therefore speculate that the sampling and retrieval biases of the OCO-2 satellite at high latitudes weakened the capability of our inversion system to constrain land fluxes over boreal regions. Since total net natural fluxes are conserved globally, flux underestimations in particular regions are typically compensated for by flux overestimations of similar magnitudes in other regions through the atmospheric inversion process, thus resulting in large variations in regional flux estimates among different inversion models.

**3.3 Seasonal cycle of carbon fluxes**

The different atmospheric inversion systems analysed herein presented broadly consistent phases (source-to-sink transitions) and amplitudes (peak-to-trough differences) of the seasonal natural land flux cycle except over the tropical region (23°S–23°N) (Fig. 6). Predominant sinks were identified over the Northern Hemisphere during the growing season, with maximum monthly sinks occurring in July (Figs. 6a and 6b). The prior flux used in our inversion system revealed a smaller carbon uptake in July (the red dashed curves in Figs. 6a and 6b); this peak is substantially enlarged (the red solid curves in Figs. 6a and 6b) in the posterior fluxes after assimilating OCO-2 XCO$_2$. During fall and winter in the Northern Hemisphere, the shift from sink to source is consistently reproduced by different atmospheric inversion models, although the satellite-based posterior fluxes tend to follow the pattern of the prior due to a lack of valid OCO-2 XCO$_2$ retrievals over the 50–90°N region (Figs. 2c and 2d). Overall, satellite-based inversions (e.g., our inversion model, CAMS v20r3, and CMS-Flux) tended to differ from the surface-based inversions (e.g., CAMS v20r2, CT2019B, and Jena) regarding the output peak sink estimates in the growing season. The satellite-based inversions estimated carbon fluxes of −1.28−−1.38 GtC month$^{-1}$ over 50–90°N in July (Fig. 6a), and these values were slightly smaller than the surface-based inversion estimates (−1.58−−1.83 GtC month$^{-1}$). Over the 23–50°N latitudinal band (Fig. 6b), the satellite-based inversions estimated larger carbon uptake magnitudes (−0.94−−1.19 GtC month$^{-1}$) in July than the surface-based inversions did (−0.82−−0.98 GtC month$^{-1}$).

The peaks and troughs identified in the carbon fluxes over the tropics (23°S–23°N) were not consistently represented by different atmospheric inversion models (Fig. 6c), indicating the need for collective efforts to improve tropical carbon budget estimates. A small seasonal cycle amplitude was revealed by our inversion results, CT2019B, and Jena, while the CMS-Flux and the two CAMS inversion models all presented relatively large seasonal cycle amplitudes. These substantial discrepancies could potentially be attributed to a lack of strong CO$_2$ observational constraints and the difficulty of accurately simulating atmospheric transport processes in the tropics. The OCO-2 satellite is expected to provide broad coverage but is greatly hindered by cloud coverage during the wet season and aerosol pollution from biomass burning during the dry season in the tropics. In addition, in previous satellite-based inversions, researchers preferred not to use OCO-2 ocean glint observations



due to known uncertainty issues (O'Dell et al., 2018), which substantially reduced the number of assimilated OCO-2
observations over the tropics. The OCO-2 V10r satellite retrievals are thought to have improved these ocean glint
observations, which were used as observational constraints in our inversion system. Over the mid- to high latitudes of the
southern atmosphere (90–23°S), where much less land is present, the natural land flux inversion estimates did not depart
largely from the priors and were highly consistent among different inversion models (Fig. 6d).

Over the 11 TransCom regions, our inversion results exhibited seasonal cycle amplitudes similar to those of the other
analysed inversion models (Fig. 7). The peak summertime drawdown of fluxes in the northern ecosystems, which represents
the deeper sinks during the growing season, is consistently constrained by different inversion systems over the North
American Boreal (Fig. 7a), North American Temperate (Fig. 7b), Eurasia Boreal (Fig. 7g), Eurasia Temperate (Fig. 7h), and
Europe (Fig. 7k). However, these regions reveal relatively large ensemble spreads in their carbon source estimates during fall
and winter due to the sparse satellite observational constraints and divergent seasonal amplitudes of the prior fluxes used in
the inversion process, which finally result in large discrepancies in the annual flux estimates. For example, our inversions
exhibited relatively small annual fluxes over the North American Boreal compared to the other inversion estimates (Fig. 5),
and this was mainly due to the larger carbon sources derived between September and February (Fig. 7a). We also observed
substantial disagreements in the seasonal cycle of the flux amplitude over the South American Tropical (Fig. 7c), Tropical
Asia (Fig. 7i), and Australia (Fig. 7j) regions; these amplitudes were found to be close to carbon neutral based on the
ensemble of different inversion models (Fig. 5) but diverged widely with regard to their annual and monthly flux estimates.

### 3.4 Evaluation with $CO_2$ measurements

The GEOS-Chem-modelled $XCO_2$ outputs based on posterior fluxes matched the OCO-2 $XCO_2$ retrievals in terms of both
their magnitudes (Fig. S1) and trends (Fig. S2), thus suggesting that our inversion system was effectively constrained by the
assimilated OCO-2 $XCO_2$ values. The modelled RMSEs of the posterior fluxes against the OCO-2 $XCO_2$ values were
constrained by our inversion system (Fig. S1). The posterior simulations (the red curves in Fig. S2) corrected the
overestimated prior-modelled $XCO_2$ values (the blue curves in Fig. S2) compared to the OCO-2 observations (black curves
in Fig. S2) by adding terrestrial carbon uptake to the prior flux. Regarding the trends and interannual variability, the
simulations driven by prior fluxes overestimated the increasing $XCO_2$ over the Northern and Southern Hemispheres, and this
was corrected in our inversion system by increasing the terrestrial carbon sinks from 2015 to 2020 (Table 3).
The atmospheric $CO_2$ concentration measurements obtained at surface (Table S1) and by aircraft (Table S2) networks both
confirmed that the $CO_2$ values modelled based on posterior fluxes (the red curves in Fig. 8) were improved relative to those
based on prior fluxes (the blue curves in Fig. 8); thus, this process substantially reduced the biases (Figs. 8a–c) and RMSEs
(Figs. 8d–f) compared to the $CO_2$ observations with regards to both latitude and altitude. The daily, seasonal, and interannual
variations in surface $CO_2$ concentrations were reproduced in the posterior flux-based simulations, as illustrated by the six
selected stations (Fig. S3) that varied by latitude and altitude and provided continuous measurement records between 2015
and 2020. The comparisons with aircraft observations in the free troposphere (above 3000 m) showed slightly smaller biases



(Fig. S4) because these measurements were less affected by local sources (Chevallier et al., 2019). The evaluations conducted at the three atmospheric layers all presented large RMSEs in the northern extratropics (Fig. 8), suggesting that the inversion estimates of $CO_2$ fluxes in this region tended to have relatively larger uncertainties than the other latitudes; this

finding was consistent with the atmospheric inversion ensemble assessment of Crowell et al. (2019). The northern extratropics are dominated by land but lack adequate, high-quality OCO-2 $XCO_2$ retrievals during fall and winter, therefore contributing weak observational constraints to the flux outputs.

## 4 Discussion

### 4.1 Influence of prior fluxes and uncertainties

The sensitivity inversion results diverged with regard to the land and oceanic fluxes (Table 4), although all inversion results agreed with the NOAA atmospheric $CO_2$ growth rates, suggesting that the prior fluxes associated with uncertainties altered the global carbon budget partitioning scheme in the atmospheric inversion results. The global net ecosystem exchange derived from our reference inversion model was $-3.48$ GtC yr$^{-1}$ in 2015; this value was substantially different from the S_exp1–S_exp4 estimates ($-2.99$–$-3.83$ GtC yr$^{-1}$). The global oceanic flux inversion estimates were adjusted accordingly in

each inversion to match the atmospheric $CO_2$ growth rates. Based on the evaluations performed using surface $CO_2$ measurements (Fig. S5), our reference inversion results presented slightly smaller biases and RMSEs in the modelled $CO_2$ over the tropics and northern latitudes than S_exp1, S_exp2, and S_exp4. The prior fluxes used in S_exp3 were derived from the CASA model (Table 2), and this experiment exhibited better performances in the northern mid-latitudes but presented larger biases and RMSEs than the reference inversion in the northern high-latitudes, possibly due to weaker constraints

associated with satellite observations and inappropriate prior fluxes used in this region. Increasing the data assimilation window length to 14 days (S_exp4) slightly increased the global NEE and decreased the oceanic fluxes (Table 4), while the inversion model performance evaluated with surface measurement of $CO_2$ concentrations are not improved (Fig. S5).

The TransCom land regions showed different sensitivities to the prior fluxes used in the atmospheric inversion process at the annual (Fig. 9) and monthly (Fig. 10) timescales. For example, the flux estimates were broadly consistent among different

inversion experiments over Eurasia Temperate and Europe, where the inversions were not as sensitive as other regions to prior information due to the stronger observational constraints of the OCO-2 satellite measurements over mid-latitude areas. The sensitivity inversion estimates of fluxes are also broadly consistent over Southern Africa, Eurasia Boreal, and Australia. In contrast, the flux estimates over the North American Boreal and North American Temperate differed greatly in their signs and magnitudes across sensitivity inversions because the atmospheric inversion system not only gave more weight to the

prior information but also represented the residual fluxes resulting from global optimization over these regions due to the weak observational constraints available in fall and winter (Figs. 10a and 10b). Large differences were also evident over the tropics (e.g., the South American Tropical, Northern Africa, and Tropical Asia) due to cloud- and aerosol-caused gaps in the satellite observations. In S_exp3, the amplitude in the seasonal cycle of carbon fluxes in the South American Tropical (Fig.





10c), Northern Africa (Fig. 10e), and Tropical Asia (Fig. 10i) regions changed substantially compared to the other inversions,
thus illustrating the large influence of prior fluxes on regional atmospheric inversions over the tropics.

## 4.2 Limitations and future perspectives

Atmospheric inversions are inherently ill-constrained due to the sparseness and uneven distribution of $CO_2$ observations;
additionally, these shortcomings are exacerbated by uncertainties in the process by which fluxes are associated with $CO_2$
concentrations in atmospheric transport model simulation. In the regions and months that lack adequate-quality observations,
the prior information tends to be given more weight when estimating fluxes through inversion. Given the global optimization
strategy of atmospheric inversions, the uncertainties associated with flux estimates over a given region can be propagated
into another region representing a residual resulting from matching global observational constraints. Our analysis suggests
that regional and monthly flux estimations are divergent across different atmospheric inversion models and even among the
results of the same model under different configurations, although these monthly flux estimates can be integrated to estimate
consistent global fluxes in line with the atmospheric growth rate of $CO_2$. Our sensitivity inversions further revealed the
considerable sensitivities of the regional inversion fluxes to the prior fluxes and their uncertainties, thus illustrating the
difficulties associated with the consistent optimization of carbon fluxes from the global to the regional scale.

Although our inversion system exhibited a good performance based on the evaluations against independent observations,
regional-scale uncertainties still exist due to the inversion model limitations discussed above. The development of an
atmospheric inversion system is a continuing effort that can benefit from developing new algorithms to improve transport
model simulations and data assimilations and from increasing $CO_2$ observation availabilities for data constraints and
evaluation processes. The future development of our inversion system will include the following two aspects. 1) We aim to
optimize the inversion system configuration, including the assimilation window and localization length, which are currently
empirically designed based on previous literature and simplified sensitivity test. A longer assimilation window or
localization length could increase the amount of observation data used to constrain local fluxes; however, an appropriate
configuration must be determined through comprehensive sensitivity experiments and evaluations, which are time-
consuming but will be considered in future work. 2) We hope to improve the regional and seasonal representativeness of the
utilized prior fluxes, especially those over regions that lack valid $CO_2$ observations (e.g., the northern high latitudes and the
tropics). Biogeochemical models that integrate process-based modules and multiple observations can be used to improve the
prior biosphere fluxes and help reduce model biases when simulating $CO_2$ over moderate to high latitudes. The
anthropogenic and biomass burning fluxes were prescribed in our inversion system, and these fluxes could be improved
based on other inversion production chains to assimilate satellite retrievals of co-emitted short-lived reactive species, such as
nitrogen dioxide (Zheng et al., 2020) and carbon monoxide (Liu et al., 2017; Zheng et al., 2021).





## 5 Conclusions

Atmospheric inversions have the potential to significantly improve our understanding of the carbon cycle at the global and regional scales given their ability to integrate both prior information and atmospheric observations. Here, we developed a Bayesian atmospheric inversion system based on the 4D-LETKF algorithm coupled with the GEOS-Chem model; this system was constrained by OCO-2 $XCO_2$ retrievals. To the best of our knowledge, this work represents the first time the 4D-LETKF algorithm was adapted to a global carbon inversion system that assimilated OCO-2 data. With this newly developed

inversion system, we inferred global gridded carbon fluxes from the latest OCO-2 V10r retrievals and investigated their magnitudes, variations, and partitioning schemes to understand the global and regional carbon budgets between 2015 and 2020. The resulting inversion-based carbon budgets agreed with the NOAA-observed $CO_2$ atmospheric growth rates and substantially improved the modelled $CO_2$ concentrations across latitudinal bands compared with the independent ground- and aircraft-based observations. Our global and regional carbon flux inversion estimates were broadly consistent with the

other state-of-the-art atmospheric inversion models and the ensemble estimates derived from GCB2021, although discrepancies were still evident in the partitioning schemes between the natural land and ocean fluxes and the amplitude of the seasonal flux cycle over the TransCom land regions; these discrepancies could be mainly attributed to the sparse observational constraints resulting from the sampling and retrieval biases of the OCO-2 satellite and the divergent prior fluxes used in different inversion systems. We further investigated the robustness of and uncertainties in our inversion results

through four sensitivity inversion tests that varied with regard to the utilized prior fluxes, applied uncertainties, and assimilation window length; the results indicated that the reference inversion results represented the optimal configuration in the current inversion framework. Additionally, our sensitivity inversions suggested that regions in which OCO-2 coverage is lacking are sensitive to the prior flux configuration, especially the tropics and northern high latitudes. The sensitivity inversion evaluations, as well as the comparisons with previous inversion models and data products, highlighted the

dedicated future development direction of our atmospheric inversion system, representing a continuous and ongoing effort.

**Data availability.**

The inversion datasets generated in this study are available from the corresponding author on reasonable request.

**Author contributions.**

BZ designed the study and wrote the paper. YK performed the atmospheric inversion experiments, carried out the analysis,

and prepared the initial draft. All of the authors provided research ideas, participated in the interpretation and discussion of the inversion results, and contributed to the writing and editing of the paper.





**Competing interests.**

The corresponding author has declared that neither they nor their coauthors have any competing interests.

**Acknowledgements.**

This work has been funded by the Young Elite Scientists Sponsorship Program by CAST (YESS20200135), the Scientific Research Start-up Funds (QD2021024C) from Tsinghua Shenzhen International Graduate School, and the National Natural Science Foundation of China (41921005). The OCO-2 retrievals were produced by the OCO-2 project at the Jet Propulsion Laboratory, California Institute of Technology, and obtained from the OCO-2 data archive maintained by the NASA Goddard Earth Science Data and Information Services Center. The authors would like to thank Ed Dlugokencky, Colm
Sweeney, Ken Schuldt, Kathryn McKain, Bianca Baier, Anna Karion, Kathryn McKain, John B. Miller, and Charles E. Miller from NOAA, Kenneth Davis from Penn State University, and Steve Wofsy, Bruce Daube, and Roisin Commane from Harvard University for providing the in situ and aircraft $CO_2$ measurement data. Additionally, we would like to thank all of the other contributors of the ObsPack data product. Andy Jacobson from NOAA and Andrew Schuh from Colorado State University provided the ObsPack diagnostic tools. The authors would also like to thank the GEOS-Chem team for providing
the GEOS-Chem transport model code and manuals.

**Financial support.**

This research was supported by the Young Elite Scientists Sponsorship Program by CAST (grant no. YESS20200135), Scientific Research Start-up Funds (grant no. QD2021024C) from Tsinghua Shenzhen International Graduate School, and the National Natural Science Foundation of China (grant no. 41921005).



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



**Table 1. Configuration of the atmospheric carbon inversion system developed in this study.**

| Model setup | Configuration | Main reference |
|---|---|---|
| **Inversion general setup** | | |
| Spatial scale | Global | / |
| Spatial resolution | 4° latitude × 5° longitude | / |
| Assimilation window | 7 days | Zhang et al. (2015), Liu et al. (2019), and Jiang et al. (2021) |
| Carbon flux optimization | The first day of each assimilation window | / |
| Horizontal localization length | 1200 km | / |
| Bayesian inversion algorithm | 4D-LETKF | Hunt et al. (2007) |
| Ensemble size | 24 | / |
| Time period | September 2014 to December 2020 | / |
| **Transport model** | | |
| Model name and version | GEOS-Chem v12.2.1 | https://geos-chem.seas.harvard.edu/ |
| Meteorological forcing | GEOS-FP | http://wiki.seas.harvard.edu/geos-chem/index.php/GEOS-FP |
| Spatial resolution | 4° latitude × 5° longitude × 47 levels | / |
| Carbon flux data | Fossil fuel: ODIAC2020 and Carbon Monitor | Oda et al. (2018) and Liu et al. (2020) |
| | Biomass burning: GFED 4.1s | van der Werf et al. (2017) |
| | Ocean flux: The $p$CO$_2$-Clim prior of CT2019B | Jacobson et al. (2020); Takahashi et al. (2009) |
| | Biosphere flux: Simple Biosphere Model, verison 4.2 | Haynes et al. (2021) |
| Initial CO$_2$ concentration field | CT2019B | Jacobson et al., (2020) |
| **Prior information** | | |
| Control vector ($\mathbf{x}^{\mathrm{b}}$) | Scale factors for daily gridded surface carbon fluxes excluding fossil fuel and biomass burning emissions | / |
| Ensemble mean ($\bar{\mathbf{x}}^{\mathbf{b}}$) | Average of three values including the optimized results from the two previous time steps and a fixed value of one | Peters et al. (2007) |
| Error covariance of $\mathbf{x}^{\mathrm{b}}$ | Normal distribution with a standard deviation of 3.0 | / |
| **Observational constraint** | | |
| Satellite observation | OCO-2 V10r bias-corrected XCO$_2$ retrievals | OCO-2 Science Team (2020) |
| Processing method | First 1-s average and then 10-s average | Crowell et al. (2019) |
| Error covariance of 10-s average | Average uncertainty of the 1-s XCO$_2$ averages, which were computed as averages of adopted individual soundings | Crowell et al. (2019) |
| Covariance inflation factor | 1.8 over lands and 1.4 over oceans | / |



**Table 2. Sensitivity inversion experiments conducted in this study.**

| Experiment | Prior terrestrial biospheric flux | Uncertainty configuration | Assimilation window | Purpose of experiment |
|---|---|---|---|---|
| S_exp1 | SiB4 model | Normal distribution with a standard deviation of 1.0 | 7 days | Analysis of the impact of smaller uncertainties on the carbon inversion results |
| S_exp2 | SiB4 model | Normal distribution with a standard deviation of 5.0 | 7 days | Analysis of the impact of larger uncertainties |
| S_exp3 | CASA model used by CT2019B | Normal distribution with a standard deviation of 3.0 | 7 days | Analysis of the impact of different prior fluxes |
| S_exp4 | SiB4 model | Normal distribution with a standard deviation of 3.0 | 14 days | Analysis of the impact of assimilation window length |





**Table 3. Global anthropogenic $CO_2$ budget from 2015–2020 derived from our reference inversion results and NOAA's atmospheric $CO_2$ growth rate.** All values are in GtC yr$^{-1}$.

| | | 2015 | 2016 | 2017 | 2018 | 2019 | 2020 | 2015–2020[d] |
|---|---|---|---|---|---|---|---|---|
| Fossil $CO_2$ emissions | | 9.63 | 9.67 | 9.79 | 10.01 | 10.06 | 9.61 | 9.80 |
| | NEE[a] | −3.48 | −3.03 | −4.30 | −4.55 | −4.03 | −4.05 | −3.91 |
| Terrestrial fluxes | Fire | 2.10 | 1.74 | 1.79 | 1.70 | 2.14 | 1.83 | 1.88 |
| | Net flux[b] | −1.39 | −1.29 | −2.51 | −2.85 | −1.89 | −2.22 | −2.02 |
| Oceanic fluxes | | −2.33 | −1.80 | −2.77 | −2.63 | −2.28 | −2.21 | −2.34 |
| Growth rate in atmospheric $CO_2$ | | 5.91 | 6.58 | 4.51 | 4.54 | 5.89 | 5.19 | 5.44 |
| NOAA $CO_2$ growth rate[c] | | 6.24 | 6.01 | 4.55 | 5.08 | 5.39 | 5.01 | 5.39 |

[a]NEE represents the net ecosystem exchange.

[b]Net flux represents the sum of the NEE and fire fluxes.

[c]NOAA $CO_2$ growth rates were obtained from https://gml.noaa.gov/ccgg/trends/gl_gr.html and were estimated based on the conversion factor of 2.124 GtC ppm$^{-1}$ (Friedlingstein et al., 2021).

[d]Annual average estimates between 2015 and 2020.





**Table 4. Global anthropogenic CO₂ budget for 2015 derived from our reference inversion results and four sensitivity inversion experiments.** All values shown here represent the global total fluxes for 2015 in GtC yr$^{-1}$.

|  |  | Reference inversion | S_exp1 | S_exp2 | S_exp3 | S_exp4 |
|---|---|---|---|---|---|---|
| Fossil CO₂ emissions |  | 9.63 | 9.63 | 9.63 | 9.63 | 9.63 |
|  | NEE | −3.48 | −3.01 | −3.59 | −2.99 | −3.83 |
| Terrestrial fluxes | Fire | 2.10 | 2.10 | 2.10 | 2.10 | 2.10 |
|  | Net flux | −1.39 | −0.92 | −1.49 | −0.90 | −1.73 |
| Oceanic fluxes |  | −2.33 | −2.63 | −2.02 | −2.48 | −1.94 |
| Growth rate in atmospheric CO₂ |  | 5.91 | 6.08 | 6.12 | 6.25 | 5.95 |
| NOAA CO₂ growth rate[c] |  | 6.24 | 6.24 | 6.24 | 6.24 | 6.24 |





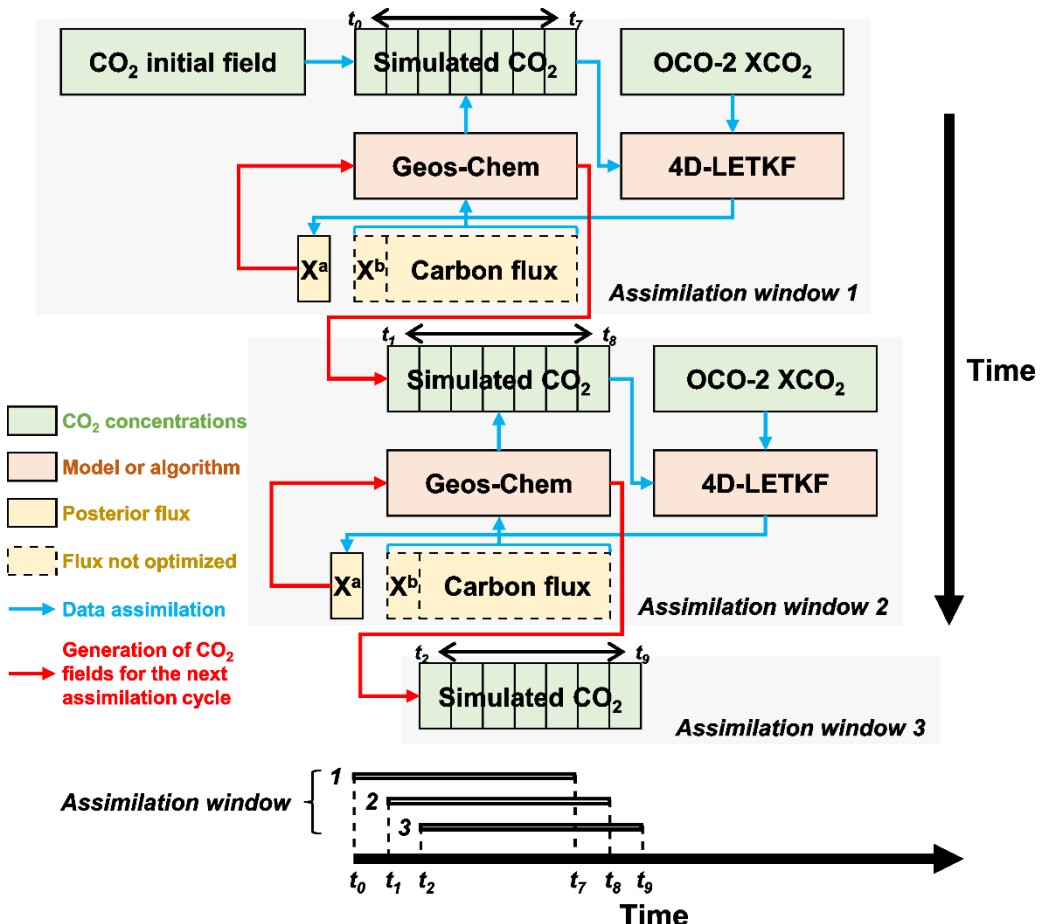

**Figure 1: Modelling structure of the carbon flux inversion system developed in this study.**




**Figure 2: Spatial and seasonal distributions of the valid OCO-2 XCO₂ retrievals between 2015 and 2020.** The numbers of days with valid OCO-2 XCO₂ retrievals (xco2_quality_flag = 0) in each GEOS-Chem 4° × 5° grid cell are shown for the periods spanning from March–May (a), from June–August (b), from September–November (c), and from December–February (d). The values shown here represent annual averages between 2015 and 2020.





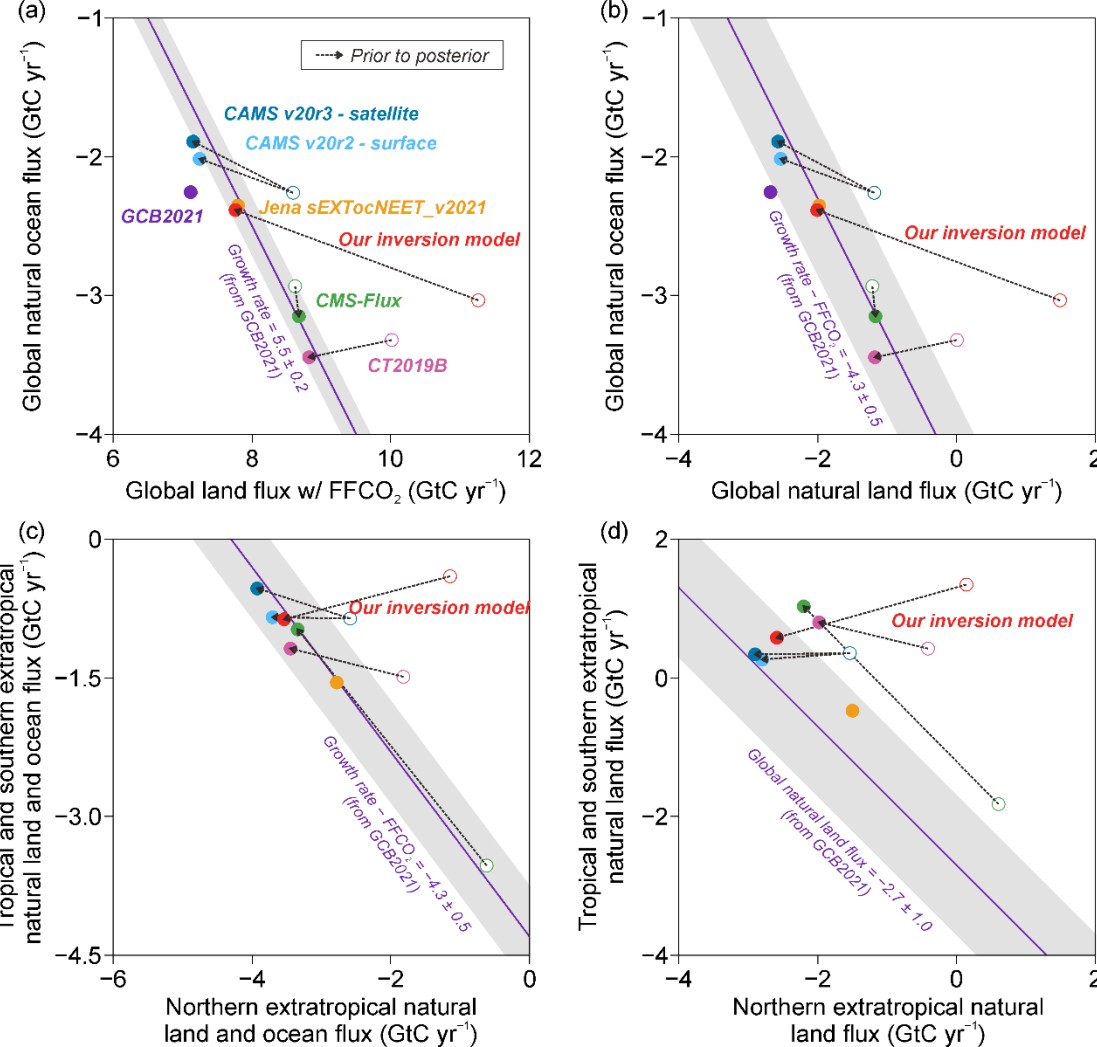


**Figure 3: Global carbon flux partitioning schemes between the land and ocean and among different latitudinal bands for 2015–2018.** The atmospheric inversion results are represented by solid circles (representing posterior fluxes) and open circles (representing prior fluxes (if any)). The red circles represent the carbon flux estimates derived from our reference inversion results; the blue circles represent the CAMS versions v20r2 and v20r3 results (Chevallier, et al., 2005); the orange circles represent the Jena CarboScope version sEXTocNEET_v2021 results (Rödenbeck et al., 2018); the green circles represent the CMS-Flux results (Liu et al., 2021); and the pink circles represent the results of the NOAA CarbonTracker version CT2019B (Jacobson et al., 2020). The purple circles in panels (a) and (b) represent the GCB2021-derived (riverine flux-adjusted) estimates (Friedlingstein et al., 2021). The purple line and equation in each panel represent the sum of the *x* and *y* variables derived from GCB2021, and the grey shaded area represents the error equivale to one standard deviation. The purple lines thus have a slope of −1, and any deviation perpendicular to these purple lines indicates disagreements in the GCB2021 estimates, including the purple circles in panels (a) and (b) derived from the GCB2021 results due to carbon budget imbalances.





**Figure 4: Spatial distribution of global natural carbon fluxes from 2015–2020.** The annual average carbon fluxes derived from 2015–2020 are shown at a spatial resolution of 4° latitude × 5° longitude. Panel (a) displays the prior terrestrial biospheric + oceanic fluxes used in the reference inversion system. Panel (b) shows the prior terrestrial biospheric + oceanic + fire fluxes (the fire fluxes were prescribed in the inversion system). Panel (c) shows the posterior terrestrial biospheric + oceanic fluxes derived from the reference inversion system. Panel (d) shows the posterior terrestrial biospheric + oceanic + fire fluxes.





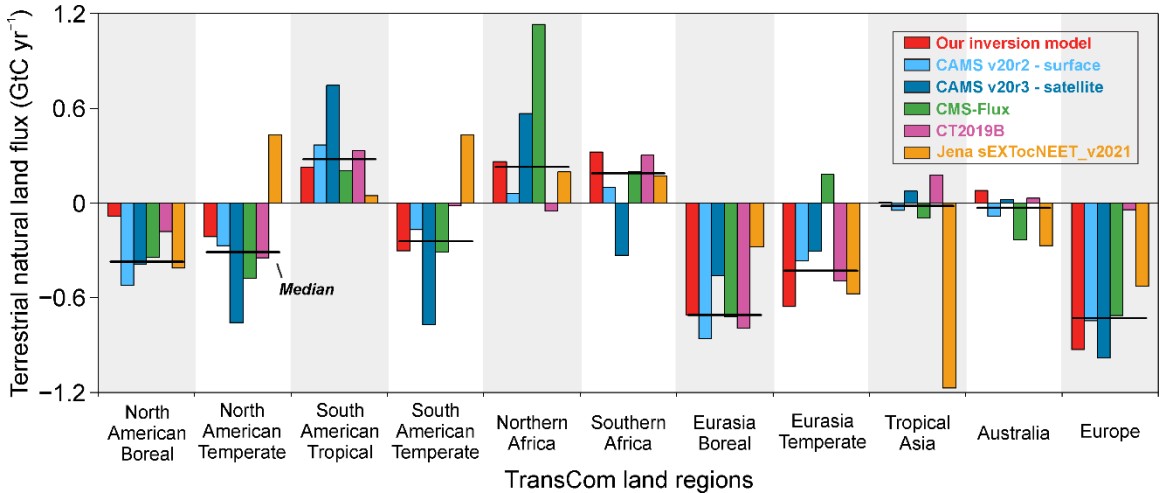

**Figure 5: Terrestrial natural land fluxes derived over the 11 TransCom land regions from 2015–2018.** Each atmospheric inversion is represented by bars showing the posterior natural land flux (i.e., terrestrial biospheric + fire fluxes) averaged between 2015 and 2018 in each TransCom land region; the black lines represent the median values of all six inversion estimates. The colours of the atmospheric inversion models are the same as those shown in Fig. 3, and the references for each inversion model are included in the caption of Fig. 3.



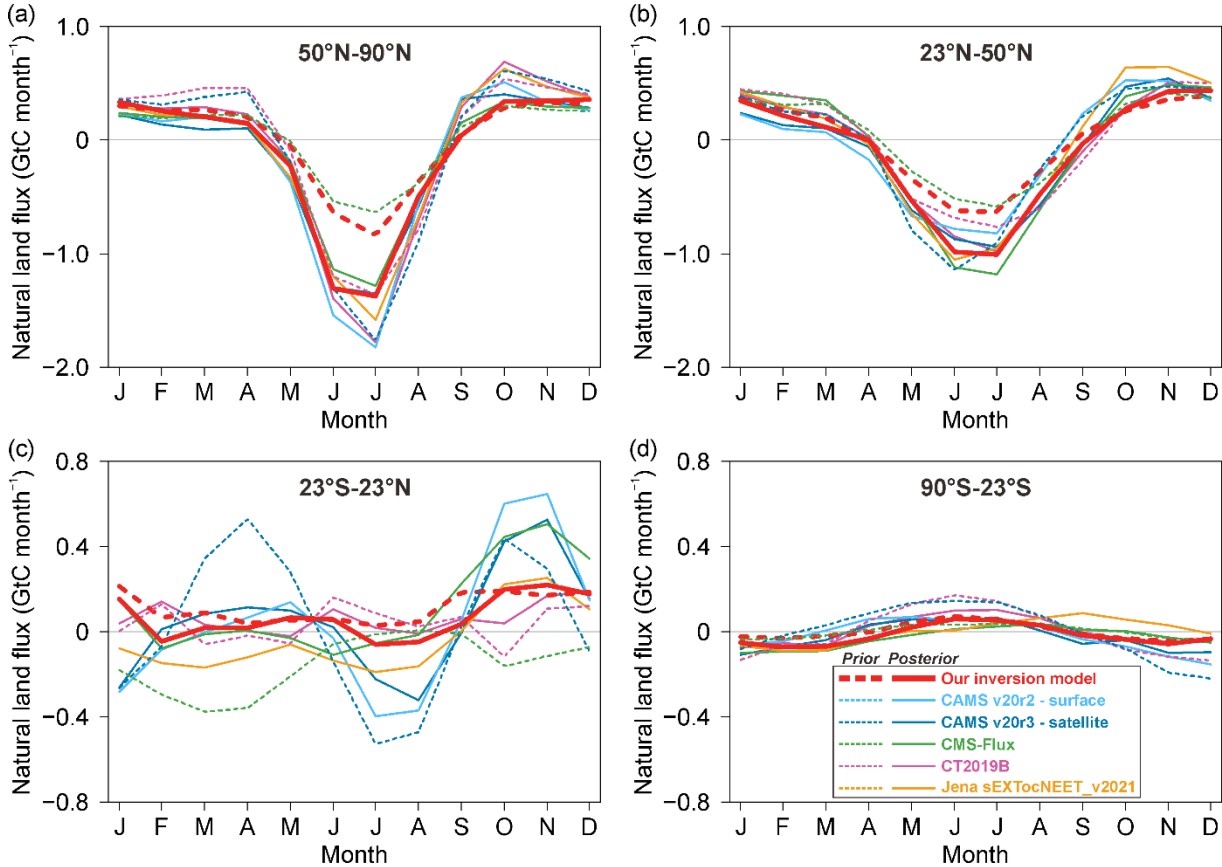

**Figure 6: Seasonal cycle amplitudes of natural land fluxes over different latitudinal bands from 2015–2018.** The global natural land fluxes (i.e., terrestrial biospheric + fire fluxes) averaged between 2015 and 2018 were split into four zonal bands: (a) the northern high latitudes (50–90°N), (b) northern mid-latitudes (23–50°N), (c) tropics (23°S–23°N), and (d) southern extratropics (90–23°S). Each atmospheric inversion result was represented by solid curves (posterior flux) and dashed circles (prior flux (if any)). The colours of the atmospheric inversion models are the same as those shown in Fig. 3, and the references for each inversion model are listed in the caption of Fig. 3.



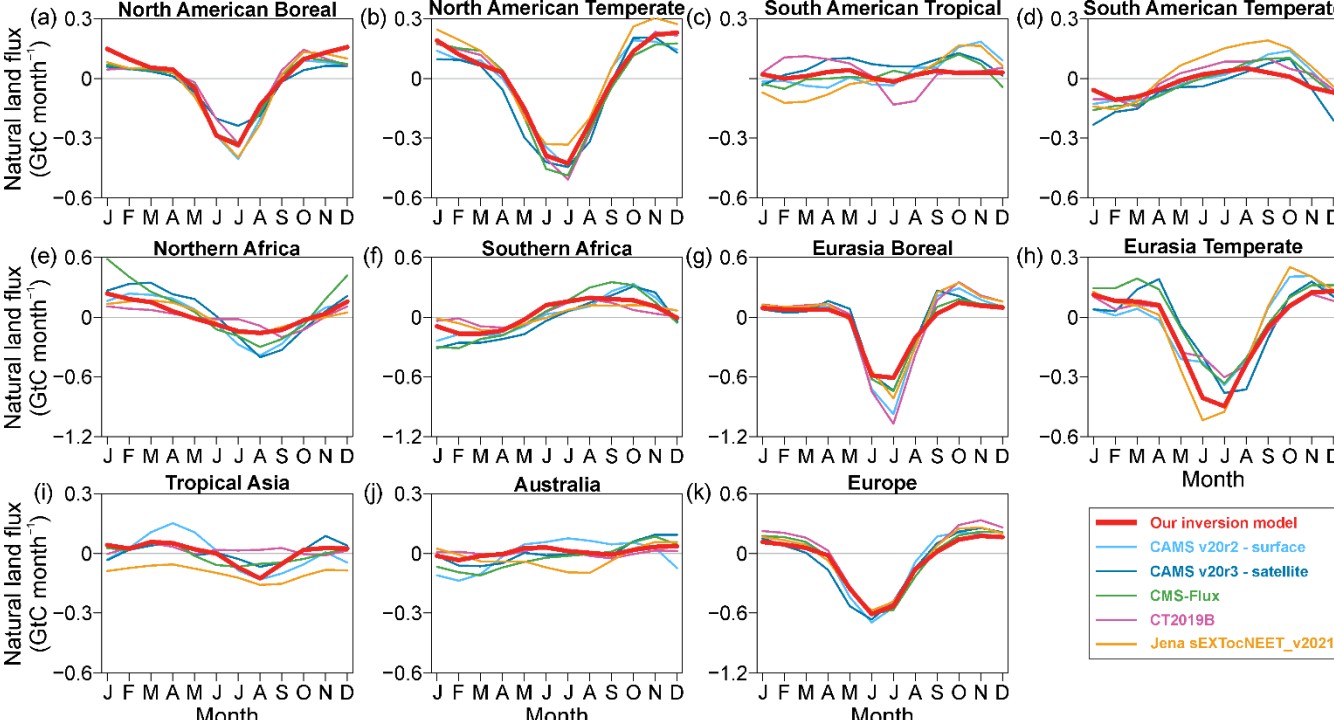

**Figure 7: Seasonal cycle amplitudes of the natural land fluxes derived over the 11 TransCom land regions from 2015–2018.** Each atmospheric inversion is represented by a solid curve representing the posterior natural land flux (i.e., terrestrial biospheric + fire fluxes) averaged between 2015 and 2018; the colours are the same as those shown in Fig. 3. The references for each inversion model are listed in the caption of Fig. 3.

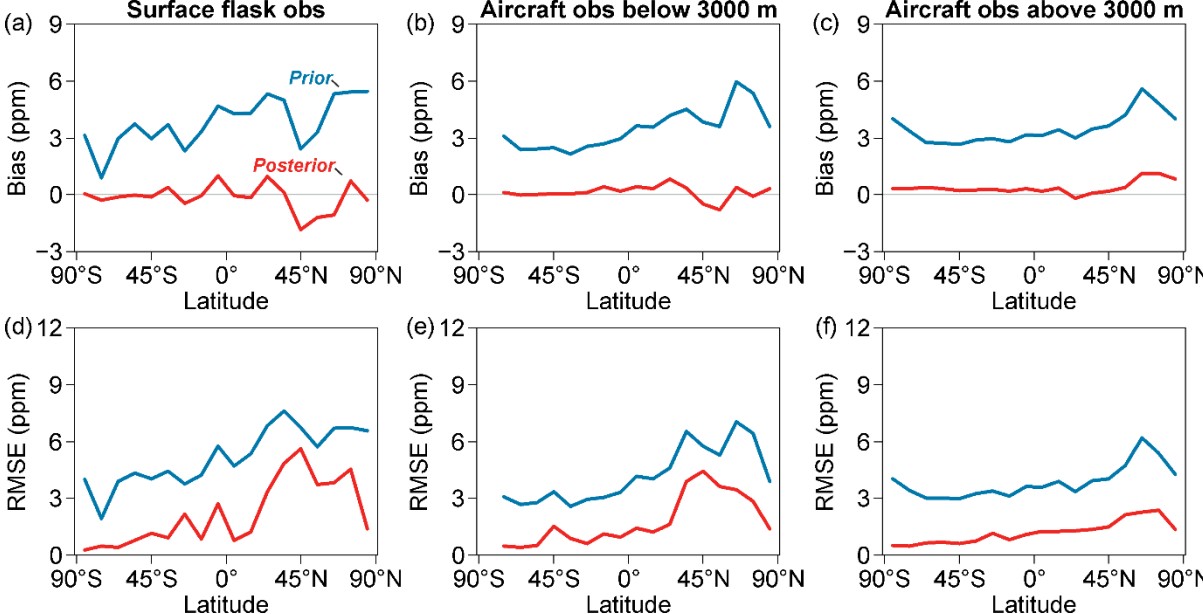

**Figure 8: Comparisons of GEOS-Chem-modelled dry air mole fractions of CO₂ with surface and aircraft measurements.** The simulations driven by the prior (blue curves) and posterior (red curves) fluxes of our reference inversions between 2015 and 2020 were evaluated against surface flask observations (a, d), aircraft observations obtained below 3000 m a.s.l. (b, e), and aircraft observations obtained above 3000 m (c, f) to derive the model biases (a–c) and RMSEs (d–f). The surface and aircraft measurement programs are summarized in Tables S1 and S2, respectively.



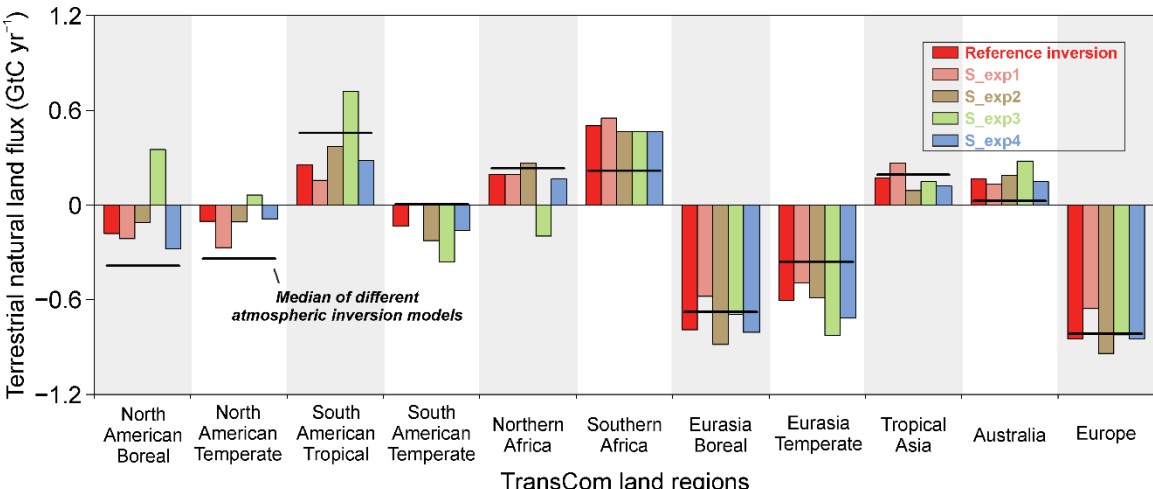

**Figure 9: Terrestrial natural land fluxes over the 11 TransCom land regions derived from sensitivity inversions for 2015.** The results of the reference inversion and four sensitivity inversions (please see Table 2) are represented by bars denoting the posterior natural land fluxes (i.e., terrestrial biospheric + fire fluxes) in 2015 for each TransCom land region; the black lines represent the median values of all six inversion estimates shown in Fig. 5 over the corresponding region and period.





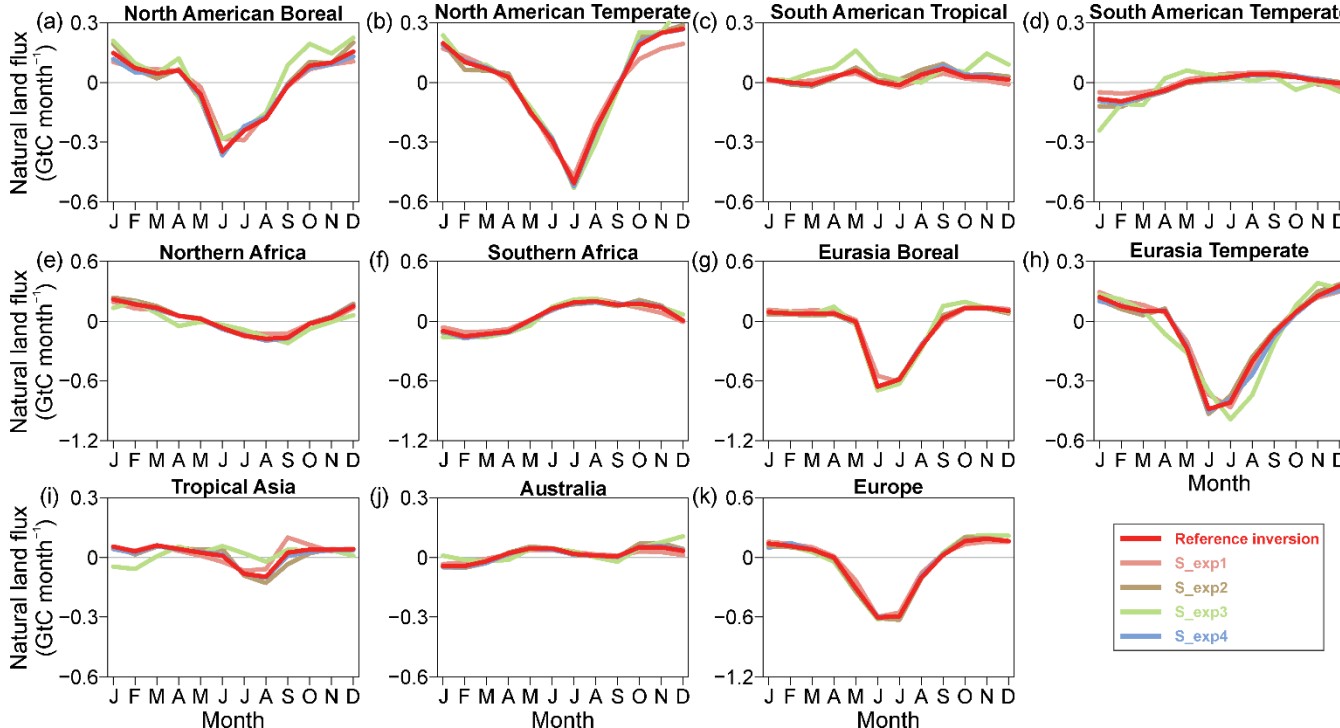

**Figure 10: Seasonal cycle amplitudes of natural land fluxes over the 11 TransCom land regions derived from sensitivity inversions for 2015.** The results of the reference inversion and four sensitivity inversions (please see Table 2) are represented by solid curves denoting the posterior natural land fluxes (i.e., terrestrial biospheric + fire fluxes) in 2015; the colours are the same as those shown in Fig. 9.