# Peer review of "Global and regional carbon budget 2015–2020 inferred from OCO-2 based on an ensemble Kalman filter coupled with GEOS-Chem"

_Atmospheric Chemistry and Physics, 2022_

## Author Comment (AC1)

**Referee #1:**

The authors present a new framework for performing atmospheric flux inversions. This framework is based on a four-dimensional local ensemble transform Kalman filter (4D-LETKF), with GEOS-Chem used as the transport model. The method is applied to version10r column-averaged $CO_2$ retrievals from the OCO-2 satellite over the five year period from 2015 to 2020. The estimated fluxes were found to be broadly consistent with those from other flux inversion systems. This represents the first application of a 4D-LETKF algorithm to perform an atmospheric inversion using OCO-2 data.

I find the paper to be well written and clear in general. The method is novel because it combines a high-dimensional grid-based parameterisation with an ensemble Kalman filtering approach. I appreciate that the authors performed sensitivity experiments to better understand what is driving the results. My main critique is that I find the mathematical description of the method lacking. My secondary critique relates to the lack of a discussion of the advantages or disadvantages of the method in comparison to other inversion systems.

**Response:**

We thank the referee for the constructive and positive comments on our paper. We provide point-by-point responses as follows.

I found it difficult to understand the method based on Section 2.1. Here are my specific questions:

How are the ensemble members initialised?

**Response:**

First, the perturbation matrix $\mathbf{X}^b$ is generated through Cholesky decomposition to the covariance matrix $\mathbf{B}$ (i.e., $\mathbf{B} = \mathbf{X}^b(\mathbf{X}^b)^T/(k-1)$) to approximate the error structure of the control vector $x^b$. Then the ensemble members $x^{b(i)}$ $\{i = 1, 2, ..., k\}$ are initialized by adding the ensemble mean $\bar{x}^b$ (i.e., calculated as the average of optimized result from the two previous time steps and a fixed value of one) to the $i$th column of the perturbation matrix $\mathbf{X}^b$. We clarified this process in our revised manuscript as follows (Lines 117-121). The color of track changes is displayed in red.

Lines 117-121:

"The prior covariance matrix $\mathbf{B}$ was constructed based on a normal distribution with the standard deviation of 3.0 within a spatial correlation length of 2000 km, and the spatial correlation of the prior flux errors between ocean and land is set to zero in our inversion. The ensemble perturbation matrix $\mathbf{X}^b$ was constructed through Cholesky decomposition to $\mathbf{B}$ (i.e., $\mathbf{B} = \mathbf{X}^b(\mathbf{X}^b)^T/(k-1)$), and the ensemble members $x^{b(i)}$ $\{i = 1, 2, ..., k\}$ were generated by adding the ensemble mean $\bar{x}^b$ to the ith column of $\mathbf{X}^b$."

The matrix B appears in equation (1) but not in equations (2)-(5). Is B the error structure for x^b that is mentioned in line 115? How does B affect the posterior state if it does not appear in the calculations?

**Response:**

Yes, $\mathbf{B}$ represents the error structure of $x^b$, which is used to initialize the ensemble members $x^{b(i)}$ and thus affects the posterior state of our inversions (Hunt et al., 2007).

What is contained within the vector x^b (and x^a, and so on)? Is it the control variables (scaling factors) for the whole assimilation period (7 days), or is it just for the first day? If it's just the first day, what are the implied values for the next 6 days, which will affect the modelled concentrations y^{b(i)}? Are these assumed to be equal to the prior mean? I looked at Figure~1 but I still could not understand what was happening.

**Response:**

The vectors $x^b$ and $x^a$ contain the scaling factors of emission fluxes for the first day of each assimilation window. For the next 6 days, these vectors adopt the prior mean $\bar{x}^b$, which represents an average from the optimized scaling factors for the two previous time steps and the current first guess value one (Peters et al., 2007). The perturbations of the modelled $CO_2$ concentrations ($y^{b(i)}$) are related to the ensemble perturbations of fluxes on the first day ($x^b$), which lays the foundation for deriving $x^a$ for the first day through equations (2)–(5). We clarified it in the revised manuscript (Lines 126-128).

Lines 126-128:

"In each assimilation cycle, the ensemble members $x^{b(i)}$ (with the ensemble mean $\bar{x}^b$ and perturbations $\mathbf{X}^b$ to approximate $\mathbf{B}$) are initialized on the first day of the assimilation window, and the following 6 days use the same $\bar{x}^b$ without perturbation."

Related to the last point, the calculation of \bar{x}^b is described as "the average optimized result from the two previous time steps and a fixed value of one". Does this calculation apply to the new day entering the assimilation period, or to all the days?

**Response:**

This calculation applies to all 7 days within each assimilation window.

The modelled concentrations y^{b(i)} must also depend on state values from before the assimilation period. Are these set to the posterior mean, or are they different from each ensemble member? What is assumed exactly?

**Response:**

The ensemble mean of posterior fluxes ($\overline{x}^a$) is used to update the carbon fluxes on the first day of each assimilation window and to drive a GEOS-Chem simulation to generate the initial $CO_2$ concentration fields for the next assimilation window (i.e., the next 7 days). The initial $CO_2$ fields are the same across different ensemble simulations ($y^{b(i)}$). We have clarified this configuration in the revised manuscript (Lines 123-124).

Lines 123-124:

"The ensemble mean of $\overline{x}^a$ is then used to update the carbon fluxes at the current day, thus driving another GEOS-Chem simulation to generate the initial $CO_2$ concentration fields for the next assimilation cycle."

I find the notation regarding \bar{x} a little confusing. Is this the unweighted average of the ensemble members? I ask because \bar{x}^a and \bar{x}^b are not unweighted averages, so the notation is a little bit inconsistent.

**Response:**

$\bar{x}$ is the unweighted average of ensemble members. $\overline{x}^a$ and $\overline{x}^b$ represent the unweighted mean of the prior ($x^a$) and posterior ($x^b$) ensemble members, respectively.

How does the localisation length work? It is stated that "y^o contains the assimilated OCO-2 $XCO_2$ within the assimilation window and localization length". Since the state vector contains every grid cell for a day, how can any observations be excluded by the localization length?

**Response:**

For the 4D-LETKF algorithm (Hunt et al., 2007), the state vector is optimized for each grid point independently. Therefore, the state vector contains only one grid for a day in each assimilation cycle, and only the OCO-2 observations within a specified distance (i.e., explicit localization length) around each grid cell are assimilated. We have added an introduction of this feature for LETKF in Lines 72-73 and Line 96.

Lines 72-73:

"In LETKF, the analysis state can be solved at each model grid independently, and only the observations within a specified local area around each model grid are assimilated."

Line 96:

"Our system assimilates OCO-2 $XCO_2$ on an ongoing basis and optimizes carbon fluxes on the first day of each assimilation window for each grid cell independently by minimizing a cost function as follows (Eq. (1))"

I think it would help for the authors to discuss how their method compares to other methods. For example, a conventional 4D-Var system has a similar state space and a similar cost function. What, in the authors view, are the advantages of their method? I think just a short discussion of the most common methods and how they compare qualitatively to the authors method would be enough.

**Response:**

Thanks for this good suggestion. We have added a short discussion in Lines 399-419.

Lines 399-419:

"The ensemble methods such as 4D-LETKF used in this study have a major advantage over the variational methods (e.g., 4D-Var) in system development simplification, but the limited ensemble size and the short spatial-temporal localization window could reduce the estimation accuracy when there is a lack of sufficient $CO_2$ observations (Chatterjee and Michalak, 2013; Liu et al., 2016). The 4D-Var method uses an adjoint model to compute the sensitivity of $CO_2$ concentrations to surface fluxes, typically associated with a long assimilation window of years (e.g., Chevallier et al., 2005; Baker et al., 2006; Liu et al., 2016), which is accurate but computationally expensive. The 4D-LETKF algorithm relates surface carbon fluxes to $CO_2$ observations through ensemble simulations upon a short assimilation window of hours to months (e.g., Kang et al., 2011; Peters et al., 2005; Bruhwiler et al., 2005). The 4D-LETKF algorithm was designed for easy implementation and computational efficiency (Hunt et al., 2007), making it easier and faster to use in high-dimensional assimilation systems than the 4D-Var method.

The explicit localization scheme in space and time for 4D-LETKF ensures the accuracy and efficiency of flux estimation based on a moderate size of ensemble members (Miyoshi and Yamane, 2007), especially over regions with sufficient observations. For example, the 4D-LETKF algorithm can achieve comparable carbon fluxes to 4D-Var over regions with dense $CO_2$ observations (Liu et al., 2016). However, over observation-sparse regions, the localization scheme of 4D-LETKF makes it difficult to optimize fluxes effectively, while the 4D-Var method can optimize carbon fluxes based on observations over a broad region where $CO_2$ concentrations are sensitive to fluxes. Increasing the duration of the assimilation window and localization length can improve 4D-LETKF performance in this case, however, impose a heavy computational burden. Alternatively, several ensemble Kalman filter studies estimated carbon fluxes for ecoregions, which reduced the system dimensions to minimize the impacts of sampling errors and the lack of observational constraints on inversions (Peters et al., 2005; Feng et al., 2009). In the future, with the increased availability of satellite $CO_2$ observations, the 4D-LETKF algorithm has the potential to play a more important role in grid-scale inversions."

Minor comments

Line 151, what does the word "integrated" mean here? Does it mean that the flux field was shifted to have annual mean zero? How was this done?

**Response:**

The word "integrated" means that the prior terrestrial biospheric fluxes are approximately equal to zero on an annual basis, although these fluxes have a seasonal cycle ( i.e., the monthly fluxes are not zero). This was done by the SiB4 model, a balanced land surface model which was designed to equate ecosystem respiration with gross primary production over one year at every grid point (Parazoo et al., 2008; Haynes et al., 2021). We have clarified this in Lines 156-157 as below.

Lines 156-157:

"We halved the gridded terrestrial biospheric fluxes to dampen the seasonal cycle and then integrated annual fluxes as zero over land based on the balance between gross primary production and respiration in the SiB4 model"

**References**

Hunt, B. R., Kostelich, E. J., and Szunyogh, I.: Efficient data assimilation for spatiotemporal chaos: A local ensemble transform Kalman filter, Physica D: Nonlinear Phenomena, 230, 112-126, doi: 10.1016/j.physd.2006.11.008, 2007.

Peters, W., Jacobson, A. R., Sweeney, C., Andrews, A. E., Conway, T. J., Masarie, K., Miller, J. B., Bruhwiler, L. M. P., Pétron, G., Hirsch, A. I., Worthy, D. E. J., van der Werf, G. R., Randerson, J. T., Wennberg, P. O., Krol, M. C., and Tans, P. P.: An atmospheric perspective on North American carbon dioxide exchange: CarbonTracker, Proc. Natl. Acad. Sci. U.S.A., 104, 18925-18930, doi: 10.1073/pnas.0708986104, 2007.

Chatterjee, A. and Michalak, A. M.: Technical Note: Comparison of ensemble Kalman filter and variational approaches for CO2 data assimilation, Atmos. Chem. Phys., 13, 11643–11660, doi: 10.5194/acp-13-11643-2013, 2013.

Chevallier, F., Fisher, M., Peylin, P., Serrar, S., Bousquet, P., Bréon, F. M., Chédin, A., and Ciais, P.: Inferring $CO_2$ sources and sinks from satellite observations: Method and application to TOVS data, J. Geophys. Res. Atmos., 110, D24309, doi: 10.1029/2005JD006390, 2005.

Baker, D. F., Doney, S. C., and Schimel, D. S.: Variational data assimilation for atmospheric $CO_2$, Tellus Series B-Chemical and Physical Meteorology, 58, 359-365, doi:10.1111/j.1600-0889.2006.00218.x, 2006.

Liu, J., Bowman, K. W., and Lee, M.: Comparison between the Local Ensemble Transform Kalman Filter (LETKF) and 4D-Var in atmospheric $CO_2$ flux inversion with the Goddard Earth Observing System-Chem model and the observation impact diagnostics from the LETKF, J. Geophys. Res.-Atmos., 121, 13066-13087, doi: 10.1002/2016JD025100, 2016.

Kang, J.-S., Kalnay, E., Liu, J., Fung, I., Miyoshi, T., and Ide, K.: "Variable localization" in an ensemble Kalman filter: application to the carbon cycle data assimilation, J. Geophys. Res., 116, D09110, doi:10.1029/2010JD014673, 2011.

Peters, W., Miller, J., Whitaker, J., Denning, S., Hirsch, A., Krol, M., Zupanski, D., Bruhwiler, L., and Tans, P.: An ensemble data assimilation system to estimate $CO_2$ surface fluxes from atmospheric trace gas observations, J. Geophys. Res., 110, D24304, doi:10.1029/2005JD006157, 2005.

Bruhwiler, L. M. P., Michalak, A. M., Peters, W., Baker, D. F., and Tans, P.: An improved Kalman Smoother for atmospheric inversions, Atmos. Chem. Phys., 5, 2691-2702, doi: 10.5194/acp-5-2691-2005, 2005.

Miyoshi, T. and Yamane, S.: Local Ensemble Transform Kalman Filtering with an AGCM at a T159/L48 Resolution, Mon. Weather Rev., 135, 3841-3861, doi:10.1175/2007MWR1873.1, 2007.

Feng, L., Palmer, P. I., Bösch, H., and Dance, S.: Estimating surface $CO_2$ fluxes from space-borne $CO_2$ dry air mole fraction observations using an ensemble Kalman Filter, Atmos. Chem. Phys., 9, 2619-2633, doi: 10.5194/acp-9-2619-2009, 2009.

Parazoo, N. C., Denning, A. S., Kawa, S. R., Corbin, K. D., Lokupitiya, R. S., and Baker, I. T.: Mechanisms for synoptic variations of atmospheric $CO_2$ in North America, South America and Europe, Atmos. Chem. Phys., 8, 7239–7254, doi: 10.5194/acp-8-7239-2008, 2008.

Haynes, K. D., Baker, I. T., and Denning, A. S.: SiB4 Modeled Global 0.5-Degree Hourly Carbon Fluxes and Productivity, 2000-2018, ORNL DAAC, Oak Ridge, Tennessee, USA., doi: 10.3334/ORNLDAAC/1847, 2021.

---

## Author Comment (AC2)

**Referee #2:**

General comments

The authors applied a flux inversion system based on LETKF algorithm to estimate the global and regional $CO_2$ fluxes based on OCO-2 satellite observations. They obtained useful results indicating ability to reconstruct the regional surface fluxes fitting within a spread of the recent global inverse modelling results by other modelling systems. On the other hand, there are deficiencies of the method that authors could not overcome and hope to improve in further developments. Paper is well written and illustrated and can be accepted after minor revisions.

**Response:**

We thank the referee for the constructive comments on our paper. We have provided our point-by-point responses as follows and revised the manuscript accordingly.

Detailed comments.

The length of optimization window of 1 week limits the power of the remote observations to constrain the fluxes. One can see a difference between fluxes retrieved with Kalman smoother when applying 1-month and 3-month assimilation window (Bruhwiler et al, 2005). The deficiency has been noted in the abstract as 'Four sensitivity experiments are performed herein to vary the prior fluxes and uncertainties in our inversion system, suggesting that regions that lack OCO-2 coverage are sensitive to the priors, especially over the tropics and high latitudes', which authors hope to address in future.

**Response:**

Thanks for your comments. We agree that the 1-week assimilation window may limit the power of remote observations to constrain fluxes. We adopt such a short window because the OCO-2 satellite provides spatially dense observations of $XCO_2$ per week over most regions, which is already sufficient to drive the 4D-LETKF algorithm to optimize fluxes. We also tested a 2-week window in the experiment S_exp4, which gave broadly consistent estimates of global and regional fluxes with the 1-week window inversions. These results have suggested that the OCO-2 provides similar constraints on fluxes despite the length of the assimilation window being doubled. However, we still observed that a few regions without sufficient OCO-2 coverage (e.g., the tropics and high latitudes) tended to be sensitive to the priors in carbon flux estimates, which could be further improved based on an assimilation window longer than 2 weeks. We will address this issue as the referee suggested in the future.

There are visible problems with posterior fluxes, as shown on Fig. 4, the range of annual mean grid fluxes occasionally goes out of reasonable range, exceeding 100 gC/m$^2$/year, pointing to a poor balance between large scale and grid scale

uncertainties, lack of a spatial correlation constraint on flux correction gradients. The results with the presumably similar algorithm in Liu et al (2019) do not show such noise, which pint to having some important differences that must be documented. Similar flux noise problem was encountered by Miyazaki et al, (2011) and later studies. Can authors isolate the cause of the problem? Could it be a result of using random grid fields as ensemble flux perturbations, while there is an alternative of using smoother random fields?

**Response:**

Thanks for pointing out this problem, which is related to the configuration of prior flux fields and their ensemble perturbations. Liu et al. (2019) did not present the noise of posterior grid fluxes like our study, probably because they sampled the ensemble perturbations based on the prior flux model instead of random grid fluxes as in our study, which better represents the spatial variations of both prior and posterior fluxes. According to our new experiment S_exp5, using a better prior flux field, we can reduce the uncertainties of prior fluxes in inversions and suppresses the noise of posterior fluxes (Fig. R1). Miyazaki et al (2011), which encountered a similar flux noise problem to our study, suggested that setting a high spatial correlation of grid flux uncertainties and doubling the ensemble size from 48 to 96 substantially reduced the noise of posterior fluxes and provided smoother posterior fields. Based on these discussions above, the methods to constrain grid-scale uncertainties include using a reasonable prior flux associated with small uncertainties, constraining the spatial correlation of grid flux uncertainties, and increasing the ensemble size.

[Figure]

**Figure R1. The global natural carbon fluxes in 2015 derived from four inversions.** The reference inversion (a), S_exp1 (b), and S_exp3 (c) are described in Tables 1 and 2 of the main text. S_exp5 (d) used the same configurations as S_exp3 except for the uncertainty of prior fluxes set as a normal distribution with standard deviation of 1.0.

Another issue related to the grid flux noise is the ensemble size. As shown by Chatterjee et al (2012), Chatterjee and Michalak, (2013) the inversion results are sensitive to ensemble size, and useful improvement are archived by increasing the ensemble size beyond 100. Miyazaki et al (2011) also obtained visible improvement of flux constraint by increasing the ensemble size from 48 to 96. Compared to those designs a system presented in this study relies on rather small ensemble size.

**Response:**

We agree that increasing the ensemble size can reduce grid flux noises. We have used an ensemble size of 24 because the LETKF performs well with a small ensemble size (e.g., Miyoshi and Yamane, 2007; Liu et al., 2019). The LETKF adopts the explicit localization schemes and the analysis is done in a much lower-dimensional space spanned by ensemble perturbations (Hunt et al., 2007). We will try larger ensemble sizes in the future development of our inversion system to constrain grid flux noises.

Despite of the visible success in weather forecast applications, LETKF use in carbon flux inversion has been tried in several studies but did not become widely used due to limitations, presumably not providing a better computational efficiency over adjoint-based variational or low rank inversion algorithms. In a revised manuscript it is advisable to mention the deficiencies of the LETKF system: limitations of small ensemble size and short window length (which may be reasonable for coupled weather-carbon cycle assimilation) and provide better arguments in support of this direction in comparison to other settings, for example Kalman filter approaches formulated by Feng et al, (2009).

**Response:**

Thanks for this good suggestion. We have added a short discussion in Lines 399-419.

Lines 399-419:

[revised manuscript text omitted]